# Dual-Modular Stems for Primary Total Hip Arthroplasty

**Jan Zajc** [1,2,*] 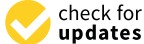 **and Samo Karel Fokter** [1,2]

1   Department of Orthopaedic Surgery, University Medical Centre Maribor, 5 Ljubljanska Street, 2000 Maribor, Slovenia; samo.fokter@ukc-mb.si

2   Faculty of Medicine, University of Maribor, Taborska Street 8, 2000 Maribor, Slovenia

\*   Correspondence: jan.zajc@ukc-mb.si

**Definition:** In primary total hip arthroplasty (THA), dual-modular stems were introduced to better restore hip stability, femoral offset, and leg length. This entry highlights the gathered knowledge about dual-modular stems and related complications in combinations with titanium (Ti) and cobalt-chrome (CoCr) exchangeable necks. The reasons for a modular neck failure are multifactorial. Some of the dual-modular stems are still on the market despite the fact th these designs have neither been proven for durability nor have shown any clinical benefits for the patients as compared to monolithic stems. Apart from very limited indications, orthopaedic surgeons should not use dual-modular stem designs for primary THA.

**Keywords:** total hip arthroplasty; hip reconstruction; titanium; cobalt-chrome; prosthesis failure; dual modularity; corrosion



## 1. History

To treat patients with advanced hip osteoarthrosis (OA), "monoblock" femoral stems were traditionally used for total hip arthroplasty (THA). These femoral stems were made of stainless steel or cobalt-chrome (CoCr) alloys allowing for the polishment of the head and were inserted with polymethyl methacrylate (PMMA) or bone cement. As the PMMA was accused off periprosthetic bone resorption and implant loosening, erroneously named "cement disease", uncemented stems of Titanium-alloys became more and more attractive. Since Ti-alloys could not be polished smoothly enough to prevent the forming of a large amount of polyethylene debris now known as the main reason for bone resorption ("polyethylene disease"); moreover, because they provided orthopaedic surgeons with better possibility to adopt leg length and femoral offset more easily, modular stems with the head as a separate piece started to appear on the market. The head was attached to the neck of the stem intraoperatively by the head-stem Morse taper. Soon, all commercially available THA systems were updated with separate modular heads, presenting with different bore lengths, diameters, and material properties. Nowadays, these "single-modular stems" present such a standard in primary THA that they are commonly referred to as the monoblock stems by the orthopaedic community to differentiate them to the later introduced dual-modular or bi-modular stems. In this manuscript, the term monoblock will be used for the standard modern femoral stem with proximal male part cylindrical Morse taper to fit a separate head.

Driven by successful utilization in revision arthroplasty, implants with dual-modular stems with an exchangeable neck emerged 30 years ago, enabling the surgeon to restore natural biomechanics of the hip joint even more closely. The exchangeable neck and femoral part of the stem are coupled together by an additional Morse taper joint, the neck-stem taper. Different neck length and orientation allow the orthopaedic surgeon to better optimise the leg length, to re-establish the joint centre of rotation, and provide better hip stability [1]. Achieving correct biomechanical conditions can theoretically slow the progress of THA components wear thus reducing particle generation now known to promote bone

resorption, ultimately leading to implant loosening necessitating revision [2]. Besides, revisions of only acetabular components were supposed to be easier with modular stem implants due to an added option of temporarily removing just the exchangeable neck and thus granting the surgeon unobstructed access to the acetabulum [2]. However, adding another junction to the system has later shown to create additional problems of metal-alloy coupling in the unfriendly environment of biological fluids that was earlier known only in marine engineering. Due to crevice corrosion, it was usually impossible to disconnect the exchangeable neck from the dual-modular stem during acetabular revision surgeries.

The GSP hip system (Cremascoli S.p.A., Milan, Italy) included one of the first commercially available stems with dual modularity intended for primary THA. Both the modular neck and the body of the stem were made of a Ti-alloy (Ti6Al4V). At the proximal end, the modular neck had a 12/14 cylindrical taper for the junction with the head. At the distal end, the modular neck had a patented 9 × 12 mm rectangular taper for the junction with the femoral stem. The stem-neck junction was extensively studied in the laboratory before the system was released and no mechanically related problems were detected [3]. The company had slightly changed the stem design after a few years, and the new system was named An.C.A. Fit. Short- and medium-term clinical results with the system were promising and persistent even in the long-term [4,5]. Later, the producer was bought by an American company (Wright Medical Technology Inc, now MicroPort Orthopaedics Inc., Arlington, TN, USA) and the design of the stem was further changed to the Profemur series, which was basically an uncemented, rectangular, grit-blasted dual-modular stem made of Ti6Al4V alloy. Despite the evolution of stem design, the design of the stem-neck junction has remained the same. Profemur hip systems were sold around the globe (Figure 1).

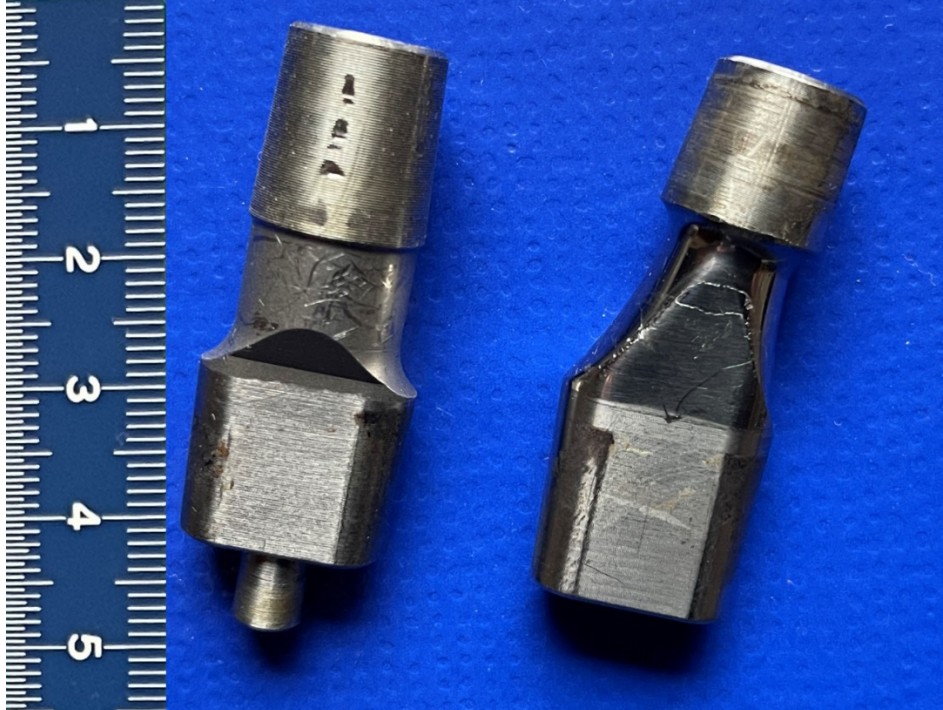

**Figure 1.** The exchangeable neck of the GSP (Cremascoli S.p.A.) (**left**) and Profemur Z (Wright Medical Technology Inc.) (**right**) dual-modular systems. Both necks are short, the GSP-neck is straight, and the Profemur Z-neck is in an 8° varus orientation. Note the same dimensions of the rectangular-shaped cross section with rounded filets of the distal Morse taper [6].

While the popularity of dual-modular system grew [7], the first reports of unexpected complications of using them for primary THA started to appear [8,9]. The producer replaced Ti-alloy necks with CoCr necks due to higher modulus of elasticity of CoCr-alloy, thus lessening the extent of micromotions [10]. However, this manoeuvre did not solve

the problem of modular neck failures. In 2015, the manufacturer (MicroPort Orthopaedics) recalled long CoCr necks of the Profemur series from the market.

Two dual-modular systems (ABG II and Rejuvenate, both from Stryker, Mahwah, NJ, USA) got recalled from the market for their increasing rates of reported complications [2,11]. After some dual modular systems were recalled, orthopaedic surgeons' enthusiasm withered [12]. Between 2017 and 2019, a 50% drop in the usage of modular stems was recorded by German Arthroplasty Register in the annual report of 2020 [13].

## 2. Types and Variations of Dual-Modular Implants

In general, stem modularity can be divided into proximal, mid-stem and distal (Figure 2) [14,15]. Head-neck junctions, anterior-posterior pads, modular collars, proximal shoulders and stem sleeves are all considered as proximal modularity [16]. As mid-stem and distal modularity became more obsolete for primary THA, proximally modular stems still remained in use for primary THA, and they will be the main focus of this paper.

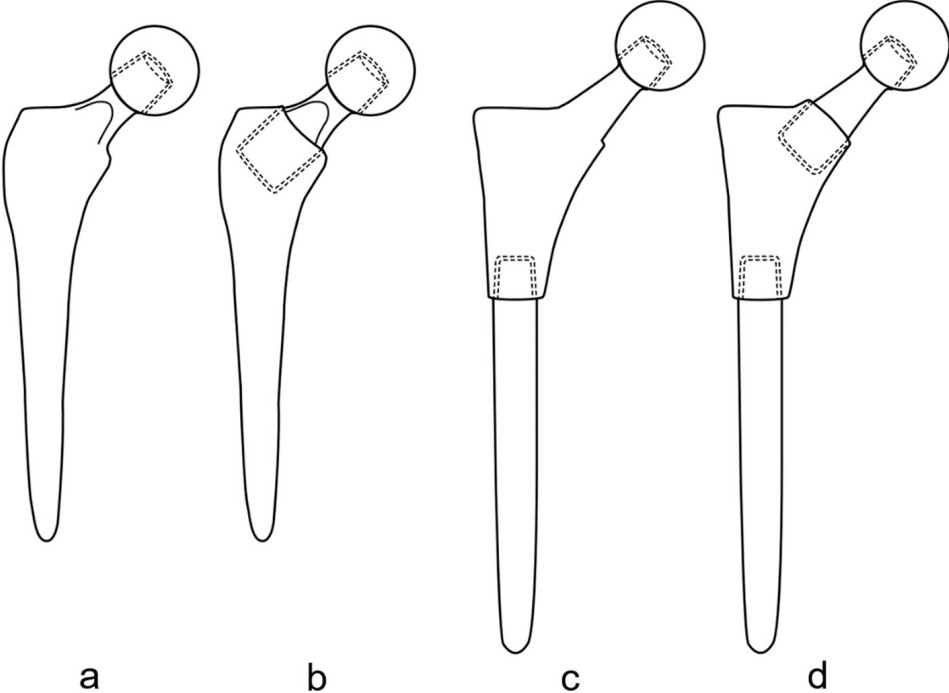

**Figure 2.** Different designs of hip stem modularity; (**a**) single-modular stem, (**b**) proximal dual-modularity, (**c**) mid-stem dual-modularity, (**d**) proximal and mid-stem modularity.

## 3. Examination and Diagnostics

Described below is the examination and diagnostics protocol before primary THA. There is no unified protocol for diagnostics of complications associated with dual-modular stems generally agreed upon.

An examination of a THA candidate starts when a patient walks through the door. A surgeon takes notice if a patient uses a walking cane, assesses general ease of movement and patients standing and sitting posture [17]. A surgeon continues to take a focused patient's history, addressing the time and gradualness of pain onset, history of traumatic events, history of past operative procedures, history of conservative treatment for this issue and general functionality with work and everyday activities.

A physician asks the patient to undress down to their underwear and examines the patient's ability to undress, and later at the end to dress independently [17]. The rest of the clinical examination is conducted in two parts. During the first part, the patient is examined while standing. In the second part, they are asked to lie on their back.

With the patient standing, the surgeon examines the pelvis, hips, and legs, looking for:

- Ischaemic or trophic skin changes, scars, sinuses;
- Swelling/mass (lipoma, trauma, tumour, infection, hernia);
- Muscle atrophy (especially gluteal) or hypertrophy.
- Deformity (leg length discrepancy, pes cavus, scoliosis), position and degree of rotation of the leg [17].

The physician will then, with the patient still standing, perform the Trendelenburg test. The patient is asked to walk before lying down, to assess their gait without shoes and more easily observe spinal movements [17].

With the patient lying on their back, the physician measures the leg length discrepancy [17]. Palpation of the hip joint is of limited value as compared to other joints. They offers an assessment of tenderness over the greater trochanter. An assessment of range of motion follows. A physician passively moves the patients' lower limb and assesses for hip flexion (normal 0–120°), rotation (normal 0–45° internal, 0–60° external) and abduction (normal 0–40°)/adduction (normal 0–25°) of each hip, starting with the healthier hip. While flexing the contralateral hip, an examiner also performs Thomas's test. A passive extension (normal 0–10°) is measured at the end, with the patient laying on the side [17].

Beside patient's history and clinical examination, radiographic diagnostics are performed as a part of preoperative evaluation. A patient undergoes an X-ray of the pelvis with both hips in an anteroposterior view. In patients with advanced anatomical abnormalities, as in developmental dysplasia of the hip (DDH) and post-traumatic OA, a computer tomography (CT) scan is warranted. As the hip-spine relationship in THA plays an important role, and may influence impingement, dislocation, and edge loading, it is a good practice for the CT or EOS (EOS Imaging, Biospace, Paris, France) scan of the hip to include the lumbar spine in cases of spinal instability, spondylolisthesis, spondylolysis, and previous spinal surgery [18].

## 4. Indications

### 4.1. Indications for Primary THA in General

Some consider a patient's request as the only proper indication for primary THA. However, a clinician can lean on some criteria to better inform the patient and guide them towards better treatment.

When the diagnosis of hip arthrosis is certain, pain and loss of function are the main evaluation criteria for deciding when to perform primary THA (Table 1) [19]. The suitability of the patient also largely depends on whether they have received any conservative treatment before [20].

**Table 1.** Different scenarios when a patient is suitable for primary THA.

| Pain | Loss of Function | Suitability for THA |
|---|---|---|
| Severe | Severe | Suitable |
| Severe | Moderate | Suitable |
| Severe (after conservative treatment) | Mild | Suitable |
| Severe (with no conservative treatment) | Mild | Not suitable |
| Moderate | Severe | Suitable |
| Moderate | Moderate | Uncertain |
| Moderate | Mild | Not suitable |
| Mild | Severe (after conservative treatment) | Suitable |
| Mild | Severe (with no conservative treatment) | Suitable |
| Mild | Moderate | Not suitable |
| Mild | Mild | Not suitable |

*4.2. Indications for Using Dual-Modular Stem*

Dual-modular stems have been introduced to restore the disturbed anatomy more easily and to avoid complementary surgery such as osteotomies in challenging cases, especially DDH and post-traumatic OA [21]. Besides DDH, there are also other anatomical difficulties where the use of dual-modular stems could be considered, such as coxa vara, increased medio-lateral offset, and disproportionality between a patient's femoral shaft size and muscle atrophy predisposing the patient for instability. A necessity for a complex reconstruction remains the major indication for their use in primary THA.

However, because of significant disadvantages related to an increased risk of mechanical failure, dual-modular stems should not be considered as the viable option—regardless of indication—in patients with a high BMI and young active male patients or patients with a high functional demand [4]. In any case, Co-Cr necks in combination with Ti-alloy stems should be avoided because they produce severe wear with adverse local tissue reaction [22].

## 5. Surgical Technique

*Primary Implantation*

Most of the dual-modular stems could be inserted through a posterior, lateral or anterior approach. The patient is placed on the operating table in the usual manner. After femoral head resection, the acetabulum is prepared with hemispherical reamers in a standard fashion. Most of the dual-modular stem systems are intended to be used in combination with press-fit acetabular sockets and these are implanted according to producer's instructions. An acetabular liner could be inserted at this point or later during trial reduction. Some systems were shipped with trial acetabular liners. Next, the surgeon's attention is focused on the femoral canal preparation. Rasps of different sizes are inserted in a stepwise fashion. Care is taken not to exaggerate with driven force in order to prevent proximal femoral fracture. The final rasp usually serves for trial reduction. Therefore, the rasp is uncoupled from the handle. At this point, the trial acetabular liner is inserted into the acetabular socket, and a trial exchangeable neck is inserted into the bore of the rasp according to preoperative planning. Most of the systems come with different neck lengths and orientation (CCD angles and version). A trial head is attached to the proximal male part of the exchangeable neck. Care must be taken at that step to ensure that the trial head diameter corresponds to the internal diameter of the acetabular liner. Trial reduction is performed, and the hip is tested for ROM, stability, and leg length. It is a good practice to check, with an image intensifier, if all the components are correctly seated. When the best combination is found, the hip is dislocated, and the trial components are removed. The original liner is inserted into the acetabulum and the original femoral stem is inserted into the femoral canal. At this point, trial reduction could be checked again with trial exchangeable neck and head, especially if the femoral stem is felt to be in different position than the final rasp. In any case, it is of fundamental importance that the bore of the modular stem (female part of the distal Morse taper) is perfectly clean of any debris and blood, as well as dry as possible before inserting the exchangeable neck. This is sometimes hard to achieve in clinical settings, but every effort is worthwhile to achieve this goal, as dirt can prevent optimal sitting of the taper and accelerates corrosion [23–25]. Finally, the selected head is attached to the cone (male part of the proximal Morse taper) of the exchangeable neck, and the whole construct is locked with one blow of a hammer over an inserted instrument with spherically moulded plastic for head protection in the longitudinal axis of the thigh. The impaction force can have a substantial influence on the deformation and corrosion of modular tapers. While the force of 8 kN is recommended to impact the head-neck coupling, limited guidance exists on the force of impaction with which surgeons should assemble the neck-stem coupling in bi-modular THA [26]. Frisch et al. have shown that direction of impaction influences stability of the modular interface and that suboptimal stability was achieved with impaction not directed in parallel with the taper junction longitudinal axis [27].

Historically, the standard combination of materials in primary THA at the appearance of modular implants was an ultra-high molecular weight polyethylene (UHMWPE) acetabular liner and a CoCr head creating the metal-on-polyethylene (M-o-P) articulation. However, as the modular stems were marketed as advanced regarding to standard monoblock stems, the systems were also advertised as advanced regarding the proposed articulating materials. Indeed, some modular stem systems with 3rd generation ceramic-on-ceramic (C-o-C) articulation performed well in the long term in single-centre studies [5]. However, others with metal-on-metal (M-o-M) articulation ended disastrously [12].

## 6. Risks with THA in General

Complications after THA are different among specific patient groups, depending on patient's age and sex, way of life, bone quality and associated diseases. Complications are generally divided by time of their occurrence on intraoperative, early post-operative and late post-operative. Elderly patients are more prone to perioperative non-surgical complications (cardiac, pulmonary, neurologic) while younger, more physically active patients are more likely to experience long term implant-related problems.

Possible complications in primary THA are:

- Deep vein thrombosis (DVT) [28];
- Pain;
- Stiffness;
- Luxation of artificial joint [29];
- Nerve damage [30];
- Bone damage;
- Bleeding [31];
- Periprosthetic fractures (intra-, postoperative) [32];
- Implant fracture;
- Myositis ossificans;
- Polyethylene liner wear;
- Infection;
- Aseptic loosening;
- Pulmonary embolism;
- Loss of limb;
- Death [32].

The risks are even greater in revision surgery, especially considering periprosthetic fractures, bleeding, and infection.

## 7. Complications with Dual-Modular Implants

*7.1. Classification of Complications*

Complications associated with the use of dual-modular hip implants can be, for ease of discussion, separated into mechanical and inflammatory complications. However, they are, in fact, a part of an interconnected tapestry of mechanical and chemical causes of failure and the body's immune systems response. Furthermore, mechanical complications can facilitate inflammatory ones and vice versa.

### 7.1.1. Mechanical Complications

The exchangeable neck is fixed inside the bore of the stem by a rectangular cross-section taper. As early as 2009, Kop et al. had raised suspicion that even with modern corrosion-resistant materials and modern taper designs, added modularity means a higher chance of metal ion generation, crevice corrosion, fretting corrosion and macro debris, which can facilitate periprosthetic bone loss and acetabular instability [33,34].

The worst mechanical complication for the patient is a modular neck fracture that can, in certain cases, happen as early as two years after the implantation [8,9,35,36]. The fretting on the modular junction is constantly removing the protective oxide layer that protects the Ti6Al4V alloy implants from uniform corrosion. Due to the continuous removal of the

passive layer, constant re-passivation occurs, depleting the oxygen supply in the crevice and thus locally dropping the pH. This further facilitates corrosive processes, leading to pitting corrosion and subsequently into the formation of sharp cracks [35]. These fatigue-cracks typically occur on the lateral side of the proximal end of oval Morse neck-stem taper, where the biggest tensile stresses occur during normal daily activities (Figure 3).

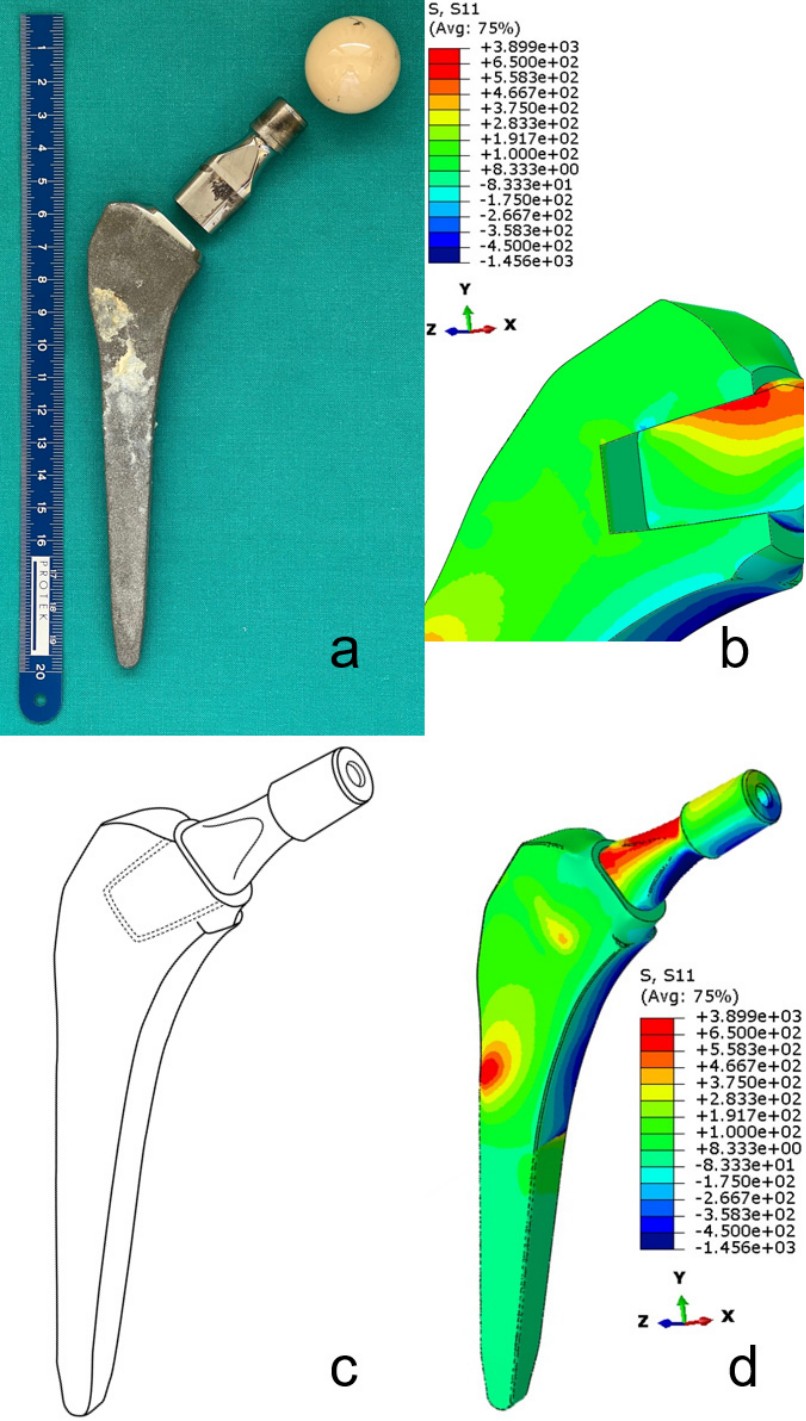

**Figure 3.** (**a**) Profemur Z modular hip system, (**b**) a close-up cross-sectional view of stress fields inside the implant when the patient stumbles and forces of 8-times bodyweight appear (presented in megapascals; S11 stress component is illustrated as it lines up with the direction of crack inducing forces), (**c**,**d**) a schematic view of Profemur Z hip implant and a representation of stress field inside the same prosthesis (in MPa).

Those tensile stresses can, while combining a long neck with an extra-long head, exceed the maximum tensile strength of the Ti6Al4V alloy, when the patient stumbles [37–39]. Sheer forces of higher magnitude appear in dual-modular systems as compared to single-modular systems with otherwise similar designs [40].

Even though cyclic external mechanical loads can initiate and propagate cracks, they are not needed for corrosion to take effect. The alloys commonly used in arthroplasty, in complete absence of mechanical influences, lose 80% of all ions that they would in total due to uniform corrosion [41]. This occurs within the first 24 h of being submerged in serum [41]. When a crack is formed beyond a certain point, self-propagating autocatalytic processes occur at the tip without the need for mechanical loads. They selectively dissolve the beta phase of the alloy. A loss of alpha-grains and their subsequent conversion into oxide then follows [42].

Beside the crack propagating from the lateral side, Gilbert et al. have noticed a crack from the medial side of the proximal end of the oval Morse taper [42]. Using backscattered electron microscopy, they have showed that crack propagation from the medial side occurs due to specific corrosion, where stress at the tip of the crack occurs due to oxide formation at the tip and not because of external loads [42] (Figure 4).

A larger head diameter, increased medio-lateral offset, a longer lever arm and a combination of dissimilar alloy metals contribute to increased taper damage found on the retrieved specimens [43]. However, beside a macroscopically apparent mechanical destruction of taper material, patient factors also contribute to clinical manifestations of complications. Factors that predispose a patient to complications are male sex, a longer lever arm, and a higher body mass index (BMI) [9,44–51].

When performing a revision surgery of the Profemur Z implant (dual-modular system) due to increasing hip pain, evidence of severe corrosion could be found on 2/3 of the retrieved specimens [52]. Evidence of advanced corrosion on neck-step taper can already be seen on extracted implants at 2.5 years after the primary THA [53]. Traces of corrosion and fretting at the stem-neck junction were observed on extracted Rejuvenate (Stryker) stems as early as 4 weeks after implantation (head-neck junctions were undamaged), which means that destructive processes at the neck-stem taper start early on, with an increasing rate over time [54]. The largest material losses in Rejuvenate (Stryker) implants at the neck-stem junction were on the medial proximal side of the neck taper [55]. Length of implantation is positively correlated with corrosion rates, but negatively with fretting rates [54].

Evidence of corrosion is not limited only to the "infamous" models known for their recall from the market. In 2019, corrosion on neck-stem taper was found on an extracted M/L Taper Kinectiv (Zimmer Biomet, Warshaw, Indiana, USA) [56]. Studies comparing recalled models and models still in use for fretting and corrosion behaviour showed corrosion of all specimens, whether recalled or not [57]. Corrosion tends to be worse in dissimilar alloy pairings [57].

Some of the differences in clinical outcomes of CoCr implants could also be due to the differences in implant microstructure [46,58]. Alloy's microstructure is not sufficiently standardised and therefore greatly varies between cast and wrought alloys, not only between different manufacturers, but between different specimens of the same hip system model as well [59]. Phase boundary corrosion, hard phase detachment and consequent metal ion release occur in alloys with carbides leading to the largest material corrosion defects [59]. When inspecting the surfaces and edges of the fractured CoCr exchangeable necks, multiple small crack nucleation sites were seen scattered on the circumference of a fracture site randomly [60]. With cyclic external loads small intergranulated areas act as local stress raisers facilitating implant failure [60].

Currently, it seems impossible to accurately determine when a neck fracture is going to occur, even if we consider all of the known predisposing factors [61]. Studies reporting on dual modular implant survival rate do not always agree. A study reporting follow-up results of 2767 hips shows a mean time to fracture of 5 years (SD ± 2.3 years) for Ti-alloy necks and only 3.4 years (SD ± 1.4 years) for CoCr necks [46]. The exchangeable neck

fractures occurred almost twice as often in the group with CoCr necks than in those with Ti-alloy necks [46].

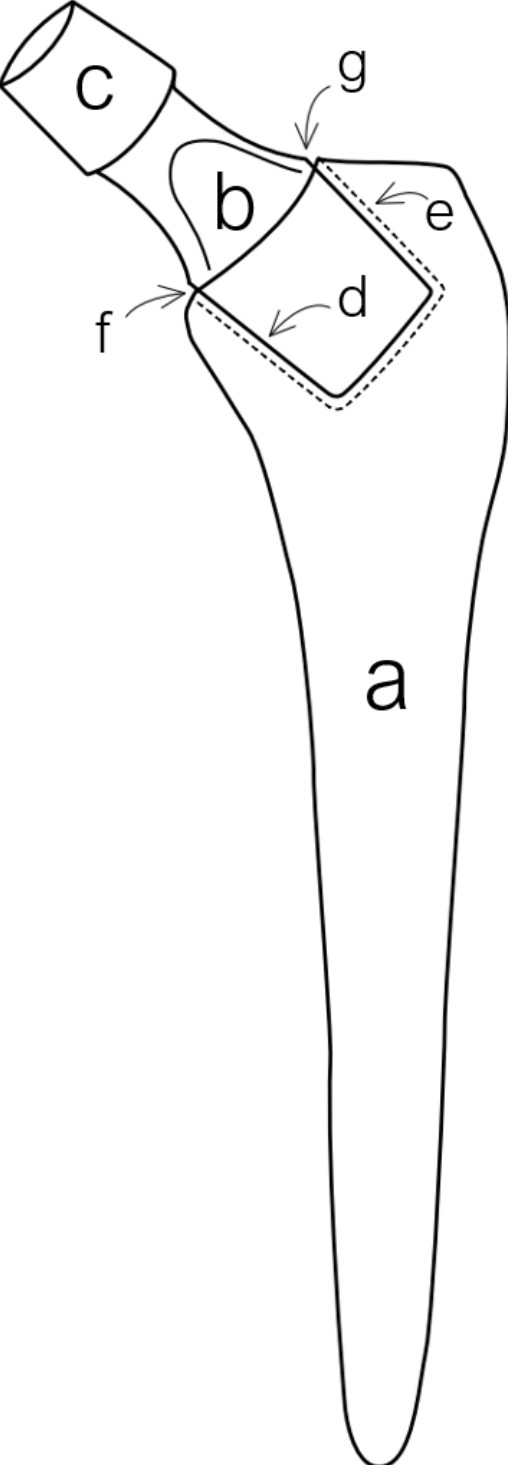

**Figure 4.** Dual-modular stem (proximal modularity); (**a**) body of stem, (**b**) neck, (**c**) neck taper at the head-neck junction, (**d**) "male" part of Morse taper, (**e**) "female" part of Morse taper, (**f**) crack propagation site without the need for mechanical load, (**g**) crack propagation site due to mechanical loads.

### 7.1.2. Inflammatory Complications

Some patients return to orthopaedic surgery clinic complaining of months of worsening pain and a decline in functionality, some as early as 6 months after primary THA [62].

Patients' symptoms are often a consequence of a pseudo-tumorous mass formed due to corrosion on the neck-stem junction in necks from CoCr alloy [63]. Increased corrosion at the neck-stem junction sometimes leads to an early-onset ALTR formation, which clinically manifested as progressive swelling and pain [62]. Similar cases are known for Ti-stem and CoCr-neck combinations. Corrosion at the interface causes a release of debris and ions, leading to large soft-tissue formations with damaged surrounding tissues [64].

Even though the patient's sex and larger offset are two of the predisposing factors for mechanical complications, where males have a greater chance of occurrence, a multivariate logistics regression analysis shows that a shorter offset is more closely connected to the formation of ALTR from corrosion along with females being more at risk [65].

Macroscopically, a high rate of soft tissue destruction results in necrotic sludge, necrotic rinds and pseudo-tumorous mass can be observed intraoperatively [66].

Microscopically, a neo-synovial proliferation with CD4 immunoglobulin and GATA-3 transcription factor rich infiltrate (both indicators of increased T-helper cell differentiation) permeating through necrosis is sometimes seen [67]. Scattered debris is found to be mixed with inflammatory cells in focal cellular areas inside a bigger necrotic area [68]. Spread of fretting corrosion products from necks made from CoCr alloy can lead to cutaneous hypersensitivity reaction, osteolysis, and aseptic lymphocyte-dominated vasculitis-associated lesion (ALVAL) with a large pseudo-tumorous mass indicating the need for revision [69].

Nano-analysis of wear particles shows needle-shaped particles that are also bigger than in other ALTR related designs, which leads to most severe soft tissue reactions out of all of the compared problematic designs [70]. Tribo-corrosion related ALTRs associated with tribo-corrosion lead to a different gene expression that is not the same as the change related with non-metal particles [71]. Co and Cr metal ions inhibit DNA-repair pathways, thus interfering with proper gene expression [2]. Continuous sequelae of such changes are effectively cancer, that can, even at concentrations considered subtoxic until now, occur in the long-term. Co ions can stimulate tumour necrotising factor alpha (TNF-$\alpha$) excretion and facilitating the apoptosis of macrophages causing macrophages and lymphocytes to die [2].

### 7.2. Diagnostics of Complications

ALTR associated with taper corrosion can clinically manifest as other more common THA related complications, for example, instability, aseptic loosening or periprosthetic joint infection [72]. There should be low diagnostic threshold to systematically evaluate patients with dual-modular implants since prompt recognition and diagnosis will speed-up the initiation of proper treatment [73]. Since in dual-modular stems metal ions are generated via different mechanisms than in MoM acetabular bearings, guidelines made for evaluation of the latter should not be used indifferently for dual-modular systems [74].

A systematic evaluation of patients with dual-modular stems should include:

- Focused patient history;
- Detailed physical examination;
- Serum metal ion levels;
- Serum inflammatory markers levels;
- Radiograph and cross-sectional imaging data;
- Hip joint aspiration tests [75,76].

Due to uncertain clinical correlations of serum ion levels, patients with dual-modular stems should undergo metal artifact reduction sequence magnetic resonance imaging (MARS MRI) scans as a part of routine check-ups [75,77].

### 7.2.1. Mechanical

To screen patients for evidence of corrosion, serum metal ion levels still appear to be the best tool [72], but rather than as an individual method, they should be used as a part of a wholesome clinical and radiographic evaluation [78]. According to national Clinical institute for clinical chemistry and biochemistry at University Medical Centre Ljubljana, Slovenia, the normal reference values for serum trace elements are less than 0.65 µg/L for

Co, less than 0.75 μg/L for Cr, and less than 6.0 μg/L for Ti. Barry et al. measured whole blood levels of Co, Cr and Ti in 36 patients with primary Profemur Preserve (MicroPort, Arlington, TX, USA) modular femoral stems with CoCr and Ti modular necks and CoC large-diameter head at a minimum 1-year follow-up [79]. The authors concluded that modular neck material had an impact on metal ion levels but the measured concentrations remained within the safe range [79]. However, Laurençon et al. found Co or Cr levels higher than 2 μg/L in 18% of patients 1 year after implantation of a dual-modular stem (Ti stem and CoCr neck) [78]. Besides, it has been shown that Ti-alloys are susceptible to early degradation and hydrogen induced effects during fretting and crevice corrosion of the interfaces contacting at the modular stem junctions in vivo [80]. We cannot predict neck fractures using serum metal ion levels as they do not significantly correlate with other fracture risk factors. They also tend to be disproportionately high in very physically active patients [81,82].

Intraarticular metal ion levels can be of great use when confirming mechanically assisted crevice corrosion (MACC) in symptomatic patients with low serum levels [83]. They are 100-times higher in affected patients than their serum values [83].

### 7.2.2. Inflammatory

We cannot correctly anticipate the amount of tissue damage found intraoperatively using pre-revision metal ion levels [84].

Adverse local tissue reactions (ALTRs) following MACC [85] usually presents in patients with bad hip stability and pain, often progressive [86]. However, the absence of pain does not rule out the presence of ALTR in patients with dual-modular implants [87].

Serum Co ion levels >2.8 μg/L and a ratio of Co- and Cr ions levels >3.8 are useful for a systematic evaluation of corrosion related ALTR, but not as a sole diagnostic parameter [88], since they tend to be elevated only in 55% of patients with pseudo-tumorous formation (ALTR after THA) [77]. There also seems to be a statistically relevant positive correlation between the size of a pseudo-tumour and serum Ti-ion levels [89].

### 7.3. Therapeutic Approach to Complications

Corrosion on taper neck-stem junctions is best treated with revision surgery [90]. A total of 13% of patients with a dual-modular stem with CoCr neck need revision surgery 2 years after the index surgery [91]. At 5 years after primary THA, the revision rate doubles [92].

### Revision

Only changing the exchangeable modular neck because of its fracture was envisioned as a simple and straight forward surgical procedure. Unfortunately, it is usually extremely difficult to pull out the fractured distal part of the modular neck that is firmly engaged in the bore of the femoral stem. Some producers have developed special tools, including a carbide steel drill bit, with which a hole can be drilled into the engaged proximal part of the fractured modular neck, exactly in its longitudinal axis. In the next step, the threads are cut into the borehole of the fractured modular neck and a screw is fastened into the borehole. A special puller is attached to the screw and firmly leaned to the proximal part of the femoral stem. In the last step of the procedure, the screw is tightened until the Morse coupling gives way, usually producing a loud sound. These tools are globally available 24/7 and are claimed to be deliverable on request in 24 h to every major clinical setting providing revision THA service.

However, only exchanging the modular neck does not solve the problem in the long term, as the new modular neck can only postpone the undesired side effects of modularity (including a potential new modular neck fracture). A study comparing revisions of monoblock and dual-modular stems coupled with large-diameter heads, showed a need for stem extraction in all dual-modular cases due to severely damaged neck-stem junctions [93]. Therefore, the modular stem itself is usually replaced during revision.

The removal of well-fixed dual modular stem presents a challenge and increases the complexity and morbidity of revision surgery [94]. To preserve the bony envelope Kwon et al. presented a "top-out" technique as an advanced alternative to conventional osteotomies [95]. Besides "top-out", there are other advanced femoral revision techniques (e.g., dual chisel technique) that could be used for removal of the dual-modular stem [96]. However, it is a good practice to lose maximally 30 min in trying to remove the stem without formal osteotomy and the surgeon should have a low threshold to perform a femoral osteotomy with stem debonding or an extended trochanteric osteotomy (ETO), if the removal of the implant proves difficult and sufficiently time consuming [7].

Co and Cr levels, C-reactive protein (CRP) levels, and erythrocyte sedimentation rate (ESR), significantly drop in the weeks following the removal of dual-modular implants [97]. Even though after the revision patients perform better on physical pain and functionality scores, an apparent drop in mental component scores is evident on SF-12 scoring system [98].

## 8. Alternatives to Dual-Modular Implants

### 8.1. "Monoblock" Systems with a Wide Range of Options

Most modern monoblock THA stems are delivered in standard and lateralised (high offset) forms as well as in a standard (135°) CCD angle and a smaller CCD angle (124° to 127°). These two parameters characterise the extramedullary feature of a stem, which, in most cases allows the surgeon to reconstruct the anatomy adequately, and reduce the need of a modular neck [99].

### 8.2. 3D CAD Scan and Additive Manufacturing

A growing trend toward the personalization of medical implants persists in the orthopaedic community [100]. In theory, custom implant designs based on patient-specific anatomy and produced by additive manufacturing can offer the patient the best available implant solution [101]. However, such implants are used for very limited indications and long-term results are lacking [102].

### 8.3. Evolving Modular Systems

In contrast with current dual-modular designs, novel modular stem-neck interfaces are being researched, designed, and tested with a goal to decrease implant breakage and enable easier intra-operative detachment of components [103].

## 9. Patient Rehabilitation

WHO defines rehabilitation as: "a set of interventions designed to optimize functioning and reduce disability in individuals with health conditions in interaction with their environment". It is a progressive, dynamic, goal- and, often, time-oriented process, that enables an individual with disabilities to recognize and attain the best mental, physical, cognitive and/or social functional level possible for him/her.

Goals of successful patient rehabilitation considering THA are as follows:

- Painless motion on the operated side;
- Independent patient mobility without any gait dysfunction;
- Functional patient independence with daily activities.

Physiotherapeutic procedures should start before the surgical procedure and continue after surgery. However, some patients with hip arthrosis present too late in their disease, with their condition so advanced, that they cannot perform physiotherapeutic exercises.

### 9.1. Pre-Operative Procedures

Physiotherapeutic exercises before the surgical treatment should facilitate better joint immobilisation and less micromovement in the joint, due to an increase in the stability of the muscular and tendinous apparatus [104].

A meta-analysis from 2019 shows that aerobic exercise is most beneficial for pain and performance [105]. Equally good for pain management is mind-body exercise (yoga, tai-chi, . . . ), but they appear to be even more beneficial in gaining function. Strengthening and flexibility/skill exercises can be used for multiple outcomes [105]. Mostly home-based exercise training programmes are usually not sufficient in reducing pain [106].

*9.2. Post-Operative Procedures*

Post-operative rehabilitation is divided into early and late.

9.2.1. Rehabilitation after Primary THA

In contrast with 50 years ago, when early rehabilitation would start a few days after THA, nowadays it begins on the first post-operative day, and it includes:

- The patient positioning properly in bed;
- Respiratory exercises;
- Learning to sit- and stand-up;
- Kinesiotherapy;
- Learning to walk with a technical aid;
- Learning to climb the stairs;
- Learning how to perform daily activities;
- Measuring Range of motion and limb length (consider need for leg raise);
- Learning of precautionary measures [107].

The day after the operation, a physiotherapist helps the patient to achieve a sedentary and then standing position. The main goal of post-operative rehabilitation is to achieve independence of motion. Pain serves as the main guide; therefore, we do not promote exercises that provoke a painful response or discomfort in patients.

Each aspect of rehabilitation mentioned above has its own specifics. In general, a patient should lie on his/her back, with an operated lower extremity supported in a slight abduction [107]. The patient performs selected respiratory exercises that can be performed in bed. He/she also performs aid-assisted respiratory exercises and inhalations if indicated by a clinician. When learning how to stand up, the patient needs to be haemodynamically stable. A physiotherapist teaches the patient to always sit over the edge of the bed on the operated side.

On the first day after the surgery, the patient begins with active and passive mobilisation. He/she actively performs different isometric and isotonic contractions as instructed by the physiotherapist. While learning to walk, a patient helps him-/herself with the use of a reciprocal walking frame at first and then transitions to using forearm crutches. The patient is taught partial weight-bearing three-point gait.

During hospitalisation, the patient must receive instructions on how to perform daily activities and how to rearrange his/her home and workplace conditions to suit his/her new needs [107]. This is, in some countries, performed by special occupational therapists. Most apparent leg length discrepancies can be evened out with physical rehabilitation. If a discrepancy is still too big or bothersome for the patient, he/she is given a leg raise to prevent gait problems and possible onset of related back pain [108]. To prevent early dislocation, preventive measures are put in place and the patient must respect them for 12 weeks after THA in which the incidence for dislocation drops by 95%. Patients must follow the techniques learned in the hospital for getting in and out of bed, sitting on and standing up from a chair, avoid low seats and use sitting raise on a toilet (in tall patients). The patient is forbidden to cross his/her legs while sitting and must walk with the support of a walking aid (as taught by physiotherapist) [107].

6–8 weeks after primary THA a surgeon performs a clinical and radiographic follow-up and according to his recommendation, the patient gradually loses the crutches and starts with everyday activities such as driving a car, recreational walking, swimming, having sex [109,110].

9.2.2. Rehabilitation after Revision Surgery

Rehabilitation after revision surgery is quite different from the one after primary THA. It is less aggressive and more slow-paced. It must also be patient-specific, since the intra-operative need for osteotomy further complicates and prolongs the rehabilitation.

Following revision surgery, weight bearing is usually restricted to toe-touch weight bearing (<5 kg of pressure on the foot with gait) for 8 weeks [111]. The patient gradually advances to partial weight bearing gait (approx. 20 kg of pressure on the foot) and then to full bearing gait unit 12 weeks postoperatively as tolerated. For more complex reconstructions, partial weight bearing gait is used until 12 weeks postoperatively, followed by slight progression to full bearing gait as tolerated by 20 weeks postoperatively. Muscle strengthening is usually avoided during the first 8–12 weeks after revision surgery. Abductor strength can take months to rebuild. To avoid dislocation after revision, the patient does not undergo hip motion exercises. An external orthosis can be used to further restrict hip motion [111].

## 10. Conclusions

While probably still a viable therapeutic choice for obsolete orthopaedic indications, dual-modular stems for primary implantation should be considered as an innovation trap [22,112]. Exchangeable necks made from Ti-alloys are susceptible to pitting, crevice, and fretting corrosion, which are precipitated by a loss of a protective oxide layer due to micromotions. CoCr exchangeable necks made from are more rigid and therefore do not produce as much micromovements [10], but added galvanic corrosion negates the benefits of promoting lesser oxide loss, leading them to fracture even earlier than Ti-alloy necks [113]. Besides sudden catastrophic failures, dual-modular Ti-alloy stems with CoCr exchangeable necks have also turned out to be inferior regarding to ALTRs [113], produce carcinogenic and otherwise toxic debris and have performed worse than in combination with Ti-alloy necks in the long-term [14,51].

Morphological mechanism of evidence of soft tissue destruction found on histology cannot be solely contributed to higher local metal ion concentration and is most likely a multifactorial display of complex interplay between patient- and implant-related factors [84,114].

Early and active rehabilitation after primary THA enables faster patient recovery. However, rehabilitation after revision surgeries, as mentioned above, is much more challenging for the patient and for the therapist.

We do not currently have an efficient diagnostic tool to predict oncoming exchangeable neck fractures. Serum metal ion levels do not correlate with other known predisposing factors for neck fracture very well. Fractures can appear as many as 17 years apart in both hips in the same patient with the same predisposing factors for both hips [61].

There is also no unified prescribed protocol of following and treatment of patients with complications after dual-modular stem implantation. Clinical evaluation, serum metal ion levels, CT scans, and MARS MRI should all be taken in consideration when assessing a patient with complications after dual-modular stem implantation.

Due to risk of wear and fracture, the use of dual-modular stems should be avoided in overweight male patients with high functional demand. Since associated with accelerated corrosion, CoCr necks are totally inappropriate for use in combination with Ti-alloy stems. Long- and varus-oriented Ti-alloy necks should be used with caution because of hydrogen embrittlement in Ti6Al4V/Ti6Al4V modular taper interfaces. Potential advantages of using currently available dual-modular stems for primary THA do not outweigh the related drawbacks. However, new dual-modular stems have been engineered and tested for durability, so this may change in the future.

**Author Contributions:** Conceptualization, S.K.F. and J.Z.; Methodology, J.Z.; Validation, S.K.F. and J.Z.; Formal Analysis, J.Z.; Investigation, J.Z.; Resources, J.Z.; Data Curation, S.K.F.; Writing—Original Draft Preparation, J.Z.; Writing—Review & Editing, S.K.F.; Supervision, S.K.F.; Final approval of the manuscript, S.K.F. and J.Z. All authors have read and agreed to the published version of the manuscript.

**Funding:** This research received no external funding.

**Conflicts of Interest:** The authors declare no conflict of interest.

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
