# Peer review of "Dual-Modular Stems for Primary Total Hip Arthroplasty"

_encyclopedia, doi:10.3390/encyclopedia2020059_

Round 1
Reviewer 1 Report
In this paper, authors explained the lack of the knowledge of dual-modular stems in primary total hip arthroplasty. Especially, they touch and describe related issues of this system in combinations with titanium (Ti) and cobalt-chrome (CoCr) exchangeable necks. The paper is well-organized; however, it can be published after minor revisions:
1) Need to add scale bar for Figures 1, 3(a).
2) Need to remove overlap between legend and stress distribution from Figure 3(b). At the same time, need to increase resolution of legend and add stress units. Why do the authors provide only S11 stress component? Please, explain. Do Figures 3(b) and 3(d) have the same legend? If yes, please add it to the both images.
3) Authors explain only description of two materials like Ti and CoCr. For this reason, I recommend to change article title and indicated these materials. Otherwise, need to add description of other materials.
Author Response
Response to Reviewer 1
In this paper, authors explained the lack of the knowledge of dual-modular stems in primary total hip arthroplasty. Especially, they touch and describe related issues of this system in combinations with titanium (Ti) and cobalt-chrome (CoCr) exchangeable necks. The paper is well-organized; however, it can be published after minor revisions.
We would like to thank the Reviewer for his time to review and comment on our manuscript.
Point 1: Need to add scale bar for Figures 1, 3(a).
Response 1: We thank the Reviewer for his comments and agree that it will contribute to better illustration of dimensions. We have added a scale bar to Figures 1 and 3(a).
Point 2: Need to remove overlap between legend and stress distribution from Figure 3(b). At the same time, need to increase resolution of legend and add stress units. Why do the authors provide only S11 stress component? Please, explain. Do Figures 3(b) and 3(d) have the same legend? If yes, please add it to the both images.
Response 2: We thank the Reviewer for his comment. We have relocated the legend on Figure 3(b) and provided it in better quality. The same legend applies to Figure 3(d), where it is also added. We cannot add units directly into the legend, but we have therefore putted them in the Figure description. S11 stress component is directed in a direction relevant for the fracture to occur (x-axis). Components S22 (y-axis) and S33 (z-axis) are perpendicular to the direction in which a fracture occurs and therefore not relevant for our illustration.
We have adapted the Figure citation to inform the reader on the points requested by the Reviewer: “Figure 3 a) Profemur Z modular hip system, b) a close-up cross-sectional view of stress fields in-side the implant when the patient stumbles and forces of 8-times bodyweight appear (presented in megapascals; S11 stress component is illustrated as it lines up with the direction of crack in-ducing forces), c) and d) a schematic view of Profemur Z hip implant and a representation of stress field inside the same prosthesis (in MPa).” (Page 8, Lines 285-289)
Point 3: Authors explain only description of two materials like Ti and CoCr. For this reason, I recommend to change article title and indicated these materials. Otherwise, need to add description of other materials.
Response 3: We thank the Reviewer for pointing this out. As the Journal requires the title should be as concise as possible,suggests within 8 words, we are sorry that article title cannot be changed.
This concludes 1st Reviewer’s comments.
Reviewer 2 Report
The paper gives a nice review concerning the use of dual-modular stems for primary THA. The study reports a general concern of implantation response on patients. In the conclusion, the author mentioned about susceptibility of Ti-alloys neck to pitting, crevice, and fretting. I suggest to cite more reference article in the Mechanical complications/Inflammatory complications to support that argument. It is also necessary to describe the tolerable level of limit for dissolve metal ions (Co, Cr, Ti) and hydrogen gas (generate during corrosion) in human body. Giving recomendation to decrease the disadvantage effect of dual-modular stems would be an advantage.
Author Response
Response to Reviewer 2
The paper gives a nice review concerning the use of dual-modular stems for primary THA. The study reports a general concern of implantation response on patients.
We would like to thank the Reviewer for his time to review and comment on our manuscript.
Point 1: In the conclusion, the author mentioned about susceptibility of Ti-alloys neck to pitting, crevice, and fretting. I suggest to cite more reference article in the Mechanical complications/Inflammatory complications to support that argument.
Response 1: We would like to thank the Reviewer for this insightful comment. More references regarding susceptibility of Ti-alloys to corrosion were cited:
Gilbert JL, Buckley CA, Jacobs JJ. In vivo corrosion of modular hip prosthesis components in mixed and similar metal combinations. The effect of crevice, stress, motion, and alloy coupling. J Biomed Mater Res. 1993 Dec;27(12):1533-44. doi: 10.1002/jbm.820271210.
Meftah M, Haleem AM, Burn MB, Smith KM, Incavo SJ. Early corrosion-related failure of the rejuvenate modular total hip replacement. J Bone Joint Surg Am. 2014 Mar 19;96(6):481-7. doi: 10.2106/JBJS.M.00979.
Point 2: It is also necessary to describe the tolerable level of limit for dissolve metal ions (Co, Cr, Ti) and hydrogen gas (generate during corrosion) in human body.
Response 2: We thank the Reviewer for pointimg this out. We have added an additional paragraph in the Complications section which reads as follows:
“According to national Clinical institute for clinical chemistry and biochemistry at University Medical Centre Ljubljana, Slovenia, the normal reference values for serum trace elements are less than 0.65 μg/L for Co, less than 0.75 μg/L for Cr, and less than 6.0 μg/L for Ti. Barry et al. measured whole blood levels of Co, Cr and Ti in 36 patients with primary Profemur Preserve (MicroPort, Arlington, USA) modular femoral stems with CoCr and Ti modular necks and CoC large-diameter head at a minimum 1-year follow-up (Barry, 2017). The authors concluded that modular neck material had an impact on metal ion levels but the measured concentrations remained within the safe range (Barry, 2017). Hovewer, Laurençon et al. found Co or Cr levels higher than 2 μg/L in 18 % of patients 1 year after implantation of a dual-modular stem (Ti stem and CoCr neck) (REF Laurençon, 2016). Besides, it has been shown that Ti-alloys are susceptible to early degradation and hydrogen induced effects during fretting and crevice corrosion of the interfaces contacting at the modular stem junctions in vivo (Rodrigues, 2009).” (Page 12, Lines 415-422; Page 13, Lines 423-427)
Laurençon J, Augsburger M, Faouzi M, Becce F, Hassani H, Rüdiger HA. Systemic Metal Ion Levels in Patients With Modular-Neck Stems: A Prospective Cohort Study. J Arthroplasty. 2016 Aug;31(8):1750-5. doi: 10.1016/j.arth.2016.01.030.
Barry J, Kiss MO, Massé V, Lavigne M, Matta J, Vendittoli PA. Effect of Femoral Stem Modular Neck's Material on Metal Ion Release. Open Orthop J. 2017 Nov 29;11:1337-1344. doi: 10.2174/1874325001711011337.
Rodrigues DC, Urban RM, Jacobs JJ, Gilbert JL. In vivo severe corrosion and hydrogen embrittlement of retrieved modular body titanium alloy hip-implants. J Biomed Mater Res B Appl Biomater. 2009 Jan;88(1):206-19. doi: 10.1002/jbm.b.31171.
Point 3: Giving recomendation to decrease the disadvantage effect of dual-modular stems would be an advantage.
Response 3: We thank the Reviewer for pointimg this out. A paragraph of Conclusion section was rewritten and reads as follows:
“Due to risk of wear and fracture, the use of modular stems should be avoided in overweight male patients with high functional demand. Since associated with accelerated corrosion, CoCr necks are totally inappropriate for use in combination with Ti-alloy stems. Long and varus oriented Ti-alloy necks should be used with caution because of hydrogen embrittlement in Ti6Al4V/Ti6Al4V modular taper interfaces.” (Page 17, Lines 638-642)
This concludes 2nd Reviewer’s comments.
Reviewer 3 Report
In this manuscript, Jan Zajc et al. developed a dual-modular stem used for primary total hip arthroplasty to better restore hip stability, femoral offset, and leg length. However, current data can not support the hypothesis and more validations are suggested to be included, especially in-vivo demonstration or clinical evaluation. The current work mainly focuses on the design of these new dual-modular stems.
Author Response
Response to Reviewer 3
In this manuscript, Jan Zajc et al. developed a dual-modular stem used for primary total hip arthroplasty to better restore hip stability, femoral offset, and leg length. However, current data can not support the hypothesis and more validations are suggested to be included, especially in-vivo demonstration or clinical evaluation. The current work mainly focuses on the design of these new dual-modular stems.
Response: We thank the Reviewer for his time to review and comment on our manuscript. However, we would like to point out that the dual-modular stem mainly discussed in our manuscript was developed decades ago by an Italian company (Cremascoli S.p.A., Milan) and later modified by an US company (Wright Medical Technology Inc, Arlington, TN). The largest clinical evaluation of this particular dual-modular stem (Profemur Z, Wright Medical) was performed on the national level in 2019 with almost 3000 patients included (Kovač, S.; Mavčič, B.; Kotnik, M.; Levašič, V.; Sirše, M.; Fokter, S. K. What Factors Are Associated with Neck Fracture in One Commonly Used Bimodular THA Design? A Multicenter, Nationwide Study in Slovenia. Clin. Orthop. Relat. Res. 2019, 477 (6), 1324–1332.). In fact, more and more data are accumulating which does not support the benefits of dual modularity in primary total hip arthroplasty. The aim of our manuscript was to give an overview of the in-vivo behaviour of these stems.
This concludes 3rd Reviewer’s comments.
Reviewer 4 Report
Manuscript titled, "Dual-modular stems for primary total hip arthroplasty", highlights the gathered knowledge about dual-modular stems and related complications in combinations with titanium (Ti) and cobalt-chrome (CoCr) exchangeable necks. The manuscript is well written and organized. It makes an important contribution to the field.
Author Response
Response to Reviewer 4
Manuscript titled, "Dual-modular stems for primary total hip arthroplasty", highlights the gathered knowledge about dual-modular stems and related complications in combinations with titanium (Ti) and cobalt-chrome (CoCr) exchangeable necks. The manuscript is well written and organized. It makes an important contribution to the field.
We would like to thank the Reviewer for his time to review and comment on our manuscript.
Point 1: Moderate English changes required
Response 1: Language style has been revised and changed accordingly.
This concludes 4th Reviewer’s comments.
Round 2
Reviewer 1 Report
Good work!
Reviewer 2 Report
I have no further objection with the manuscript for the publication
Reviewer 3 Report
The revised version address all the previous questions. The reviewer suggests to be accepted.